©c Author(s) 2022. CC BY 4.0 License.





# Platform yaw drift in upwind floating wind turbines with single-point-mooring system and its mitigation by individual pitch control

Iñaki Sandua-Fernández[1], Felipe Vittori[1], Raquel Martín-San-Román[1,2], Irene Eguinoa[1], and José Azcona-Armendáriz[1]

[1]Wind Energy Department, Centro Nacional de Energías Renovables (CENER), Sarriguren, Spain
[2]DAVE/UPM, E.T.S.I. Aeronáutica y del Espacio, Universidad Politécnica de Madrid, Madrid, Spain

**Correspondence:** Iñaki Sandua-Fernández (isandua@cener.com)

**Abstract.** This work demonstrates the feasibility of an individual pitch control strategy based on nacelle yaw misalignment measurements to mitigate the platform yaw drift in upwind floating offshore wind turbines, which is caused by the vertical moment produced by the rotor. This moment acts on the platform yaw degree of freedom, being of great importance in systems that have low yaw stiffness. Among them, single-point-mooring platforms are one of the most important ones. During the last

years, several floating wind turbine concepts with single-point-mooring systems have been proposed, which can theoretically dispense with yaw mechanism, due to their ability to weather-vane. However, in this paper it is proven that the vertical moment overcomes the orienting ability, causing the yaw drift.

With the intention of reducing the induced yaw response of a single-point-mooring floating wind turbine, an individual pitch control strategy based on nacelle yaw misalignment is applied, which introduces a counteracting moment. The control strategy

is validated by numerical simulations using the NREL 5 MW wind turbine mounted on a single-point-mooring version of the DeepCwind OC4 floating platform, to demonstrate that it can mitigate the yaw drift and therefore maintain the wind turbine rotor aligned with the wind.

## 1 Introduction

Floating offshore wind energy has undergone a great development during the last years with the objective of unlocking the

huge wind energy resource in deep water regions (>50 m), where bottom-fixed wind turbines have important technical and economical restrictions. However, this type of energy source is still too expensive to compete against other energy sources in the energy market, and further efforts are needed to reduce costs (WindEurope, 2020). The substructure and the foundation account for more than third of the CAPEX of the whole system (Stehly et al., 2020), which means that, in order to make floating offshore wind energy more competitive in the market, these two components will require innovative developments to

achieve general weight reduction and reliability increase.

Generally, one of the less reliable subsystems of the wind turbine is the yaw system (Hansen, 1992; Pfaffel et al., 2017). This system consists basically of a large bearing in the tower top, like the one shown in Fig. 1, which rotates the rotor-



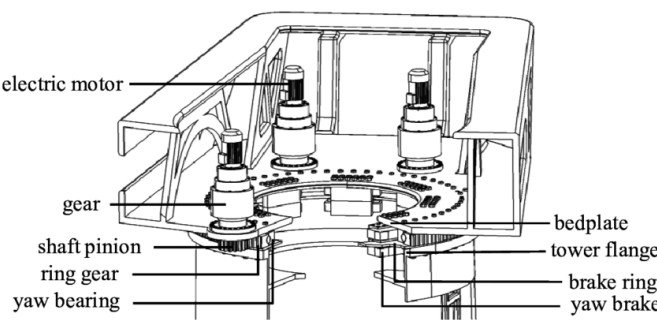

**Figure 1.** Wind turbine yaw system. Reproduced from Kim and Dalhoff (2014)

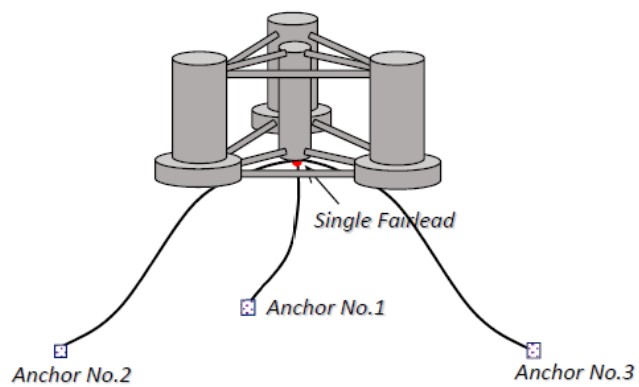

**Figure 2.** Floating platform with SPM configuration. Reproduced from Liu et al. (2018)

nacelle assembly (RNA). The yaw mechanism is responsible for maintaining the RNA aligned with the wind. Nevertheless, as this system is made of moving mechanical elements, it requires high maintenance. This drawback is especially important in
offshore environments, where O&M tasks are expensive and complicated.

In floating offshore wind turbines (FOWTs) with single-point-mooring (SPM) system, the mooring lines are attached to the platform in one single point, as shown in Fig. 2. This way, the platform can freely rotate around it and align with environmental conditions (weather-vanning). This configuration allows reducing the structural loads and potentially removing the yaw system, therefore reducing both CAPEX and OPEX of the FOWT.

The SPM system was originally conceived for ships, in order to align the vessel and reduce environmental loading caused by wind, current and waves (Chakrabarti, 2005). Nevertheless, the differences in aerodynamics between ships and FOWTs make the alignment of the latter ones much less obvious.

The alignment of wind turbines without using a yaw mechanism is a subject that has been investigated since the last century, especially in onshore downwind wind turbines (Hansen, 1992). This type of turbines offer the advantage of having a passive
yaw alignment capacity that does not require heavy yaw mechanisms. Wanke et al. (2019) explain that, when there is some



misalignment between the wind inflow and the rotor, the resulting forces on the rotor create a restorative yaw moment, which could align the rotor with the wind direction. However, for the downwind turbine analysed, there are other factors, such as the shaft tilt and wind shear, which avoid a total alignment of the rotor with the wind. This produces power losses that might make the use of a yaw mechanism compulsory again.

In the case of downwind SPM-FOWTs, it seems that a stable alignment between rotor and wind can be achieved. However, waves and currents misalignment with wind could make difficult an effective alignment of the rotor with wind, according to Urbán et al. (2021).

On the other hand, upwind turbines are the most common topology used in the wind energy sector nowadays. In onshore and offshore bottom-fixed wind turbines (using upwind configuration), the use of the yaw mechanism at tower top is needed, in order to align the rotor with the wind and maximise its power production. However, in the case of floating substructures, there is a possibility of taking advantage of the platform yaw (rotation around the vertical axis) degree-of-freedom (DoF), instead of using the yaw mechanism. Liu et al. (2018) analyse the upwind 5 MW NREL turbine supported by the DeepCWind OC4 semi-submersible with a SPM configuration (Fig. 2). Their results indicate that the FOWT has an important yaw response under co-linear wind and wave conditions, which prevent the rotor alignment with the wind. This shows that there are considerable differences in the moments generated by a downwind and an upwind rotor. Although SPM configuration helps improving the rotor orientation, it is usually not enough to keep the rotor aligned with the wind. Therefore, it is necessary to add some active system that guarantees the optimum alignment of the wind turbine. The control system seems to be adequate for this purpose, especially the individual pitch control (IPC) strategy, which is able to generate asymmetric moments in the rotor.

IPC has been traditionally applied for load reduction, based on blade-root bending moment measurements (Bossanyi, 2003). Alternatively, the usage of this strategy to improve the alignment of the wind turbine has also been tested. The IPC strategy based on nacelle yaw misalignment (known as yaw-by-IPC (Van Solingen, 2015)) has been used in onshore wind turbines, generally with downwind configuration (Van Solingen, 2015), but also with upwind one (Zhao and Stol, 2007; Navalkar et al., 2014). It has also been superficially analysed in FOWTs (Urbán et al., 2021), but only for downwind configuration, which does not take into account the challenges of controlling the alignment of upwind turbines.

The main objective of the current work is twofold: on the one hand, understand the moments that generate a platform yaw drift in upwind SPM-FOWTs and, on the other hand, demonstrate the capacity of the IPC strategy based on nacelle yaw misalignment to mitigate this drift.

To accomplish these objectives, this work is organised in the following sections. First, a description of the analysed system and its modelling is provided in Sect. 2. Then, the moment induced in yaw (or vertical) direction is explained in Sect. 3, both at rotor-level and bladed-level. This moment causes a platform yaw drift in SPM-FOWTs, which is depicted in Sect. 4. Section 5 shows a description of the IPC strategy as an alternative to mitigate the yaw drift. After it, Sect. 6 presents the resulting advantages obtained with IPC. Finally, the main conclusions and possible future working lines are presented in Sect. 7.





## 2   System description and modelling

The FOWT used in this study is the 5 MW NREL wind turbine (Jonkman et al., 2009) supported by the DeepCwind OC4
semi-submersible platform (Robertson et al., 2014), using a SPM configuration (Liu et al., 2018).

The work is carried out numerically using OpenFAST (Jonkman and Buhl, 2005), version 2.2.0. Within this tool, the floater is
modelled using HydroDyn with potential flow theory combined with Morison elements. The mooring lines are modelled using
MoorDyn, which is a lumped-mass dynamic model. In this study wave loading is not considered to show only the aerodynamic
effects. For the sake of simplicity, the tower, blades and drive train are considered rigid.

The aerodynamic model used is the in-house aerodynamic module, called AeroVIEW (**Aero**dynamic **V**ortex f**Il**am**E**nt
**W**ake), based on an implementation of a Free Vortex filament Method (FVM) (Leishman et al., 2002), combined with an
unsteady Lifting Line (LL) (Dumitrescu and Cardos, 1998) for the resolution of wake dynamics and blade loads, respectively.
This kind of aerodynamic models have been widely used in the helicopter industry (Leishman et al., 2002; Ho et al., 2017) and
are becoming more usual for offshore wind energy applications (Sebastian and Lackner, 2012; Kecskemety and McNamara,
2011). This happens because Blade Element Momentum Theory (BEMT) (Sørensen, 2016), which is the most widely aerody-
namic model used in the wind energy industry, presents limitations when predicting loads in situations with large yaw or tilt
misalignment between wind and rotor, mainly because the root vortex is not well modelled (Sant, 2007; Rahimi et al., 2016;
Gupta and Leishman, 2005). The FVM model implemented in AeroVIEW has been validated previously in yaw misaligned
conditions (Martín-San-Román et al., 2019, 2021) and allows an accurate inclusion of the effect of both the root vortex and the
blade tip vortex in both aligned and misaligned conditions.

The baseline turbine controller is an in-house development based on state-of-the-art control technologies for wind turbines.
Gain-scheduled collective pitch control is applied for generator speed control above rated. Standard IPC based on blade-root
bending moment is disabled in order to better showcase the effect of the yaw-by-IPC loop. A constant torque strategy is also
applied in the above-rated region.

## 3   Description of the yaw moment caused by the wind turbine

This section provides a description of the origin of the yaw moment generated by the wind turbine. The effects are described
at rotor level in Sect. 3.1, while the phenomena causing yaw moment at blade level are discussed and assessed in Sect. 3.2.
To improve the clarity of this discussion, the results shown in this section are performed with the onshore version of the wind
turbine introduced in Sect. 2.

### 3.1   Rotor-level description of the causes of yaw moment

One of the causes of the yaw moment produced by the wind turbine is the generator torque around the rotor shaft. When the
shaft has a certain angle with respect to the horizontal (tilt angle), this torque is projected into the vertical axis. Another effect
that generates the yaw moment is the non-symmetric aerodynamic loads caused by not perpendicular inflow winds to the rotor.

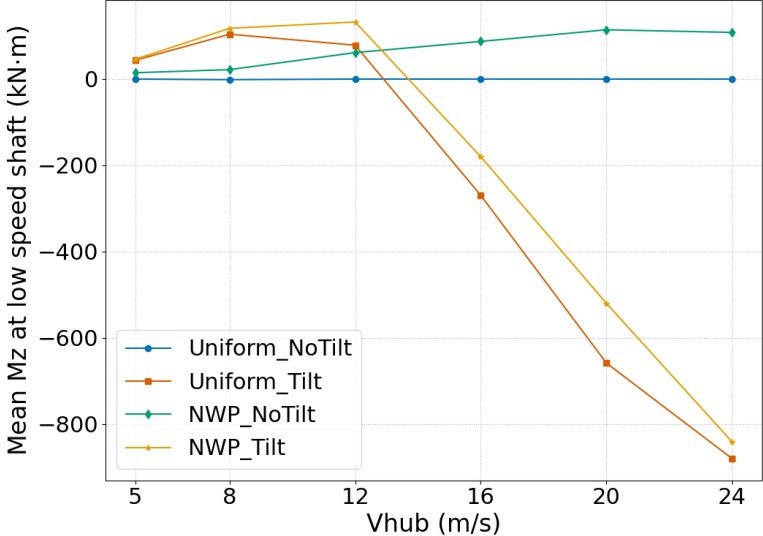

**Figure 3.** Mean aerodynamic yaw moment $M_z$ under constant wind speeds, with and without tilt angle, for uniform and NWP wind conditions (hub origin)

In this paper the only cause of not perpendicular winds is the tilt of the turbine. This tilt angle creates load variations in each of the blades as a function of the azimuthal position (as it will be explained later in Sect. 3.2) that, when they are added, result in an additional non-zero moment around the vertical axis.

In addition, the shear of the wind inflow also generates an aerodynamic imbalance in the rotor that results in a third cause of yaw moment. In this case the yaw moment appears, no matter if the turbine has tilt angle or not.

To show the influence of these effects, Fig. 3 presents the mean aerodynamic [1] yaw moment $M_z$ for a range of constant wind speeds at the rotor hub (or low speed shaft) obtained with the FVM method for the onshore 5 MW NREL wind turbine. The figure shows the results, with and without tilt angle, for two different wind conditions: under uniform wind speed, and under normal wind profile (NWP) with a exponential shear coefficient of 0.14, as defined in the guidelines (IEC, 2008).

When the shaft tilt is zero, the yaw moment is zero for the uniform wind, as there are no aerodynamic imbalances in the rotor. However, for the NWP condition, the wind shear introduces a positive moment around the vertical axis.

Both wind conditions (uniform wind and NWP) with shaft tilt show the same tendency. For wind speeds below rated (11.4 m s$^{-1}$), the aerodynamic moment in yaw is positive. Nevertheless, at above rated wind speeds, the moment becomes negative with increasing values with wind speed. At these wind speeds, the uniform wind condition produces a larger negative moment than the NWP, because the wind shear has an opposite effect on the moment.

---

[1]The moment generated in the rotor is referred herein as aerodynamic moment because it is assumed that there is no mass imbalance in the rotor.

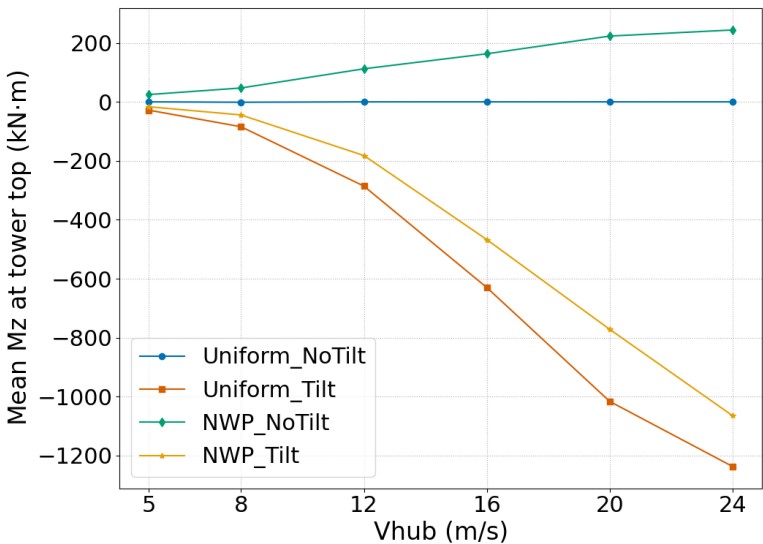

**Figure 4.** Mean total yaw moment $M_z$ (including both aerodynamic and torque projection effects) under constant wind speeds, with and without tilt angle, for uniform and NWP wind conditions (tower top)

As mentioned earlier, the total yaw moment transmitted from the hub to the tower top has an additional tilt-related component that comes from the generator torque projection in the vertical direction. Figure 4 shows the total mean yaw moment $M_z$ at the tower top, which is the sum of the aerodynamic yaw moment from Fig. 3 and the respective generator torque projection.

Again, when the shaft tilt is zero, there is no yaw moment for the uniform wind, as there is no projection of the generator torque on the vertical axis and there are no aerodynamic imbalances in the rotor. The yaw moments for NWP with no shaft tilt maintain the same tendency with respect to Fig. 3, but present larger magnitudes at the tower top, due to the horizontal distance between the hub and the tower axis. Conversely to Fig. 3, when the moment from the generator is included, the yaw moment is negative for all wind speeds for both wind conditions (uniform wind and NWP) including shaft tilt. This means, first, that the torque projection at below rated wind speeds is larger than the aerodynamic yaw moment, producing a net negative moment. Second, at above rated wind speeds, the generator torque contribution is added up to the aerodynamic moment and results in a larger negative yaw moment at the tower top.

## 3.2 Blade-level analysis of the causes of yaw moment

This section aims to provide a better insight of the causes of yaw moment at blade level, and which is their impact when the contribution of the three blades are added.

A detailed description of the effects causing the yaw moment at blade level is provided by Hansen (1992), which shows that there are four main moment contributions from each blade, namely: out-of-plane force in the wind direction, in-plane force





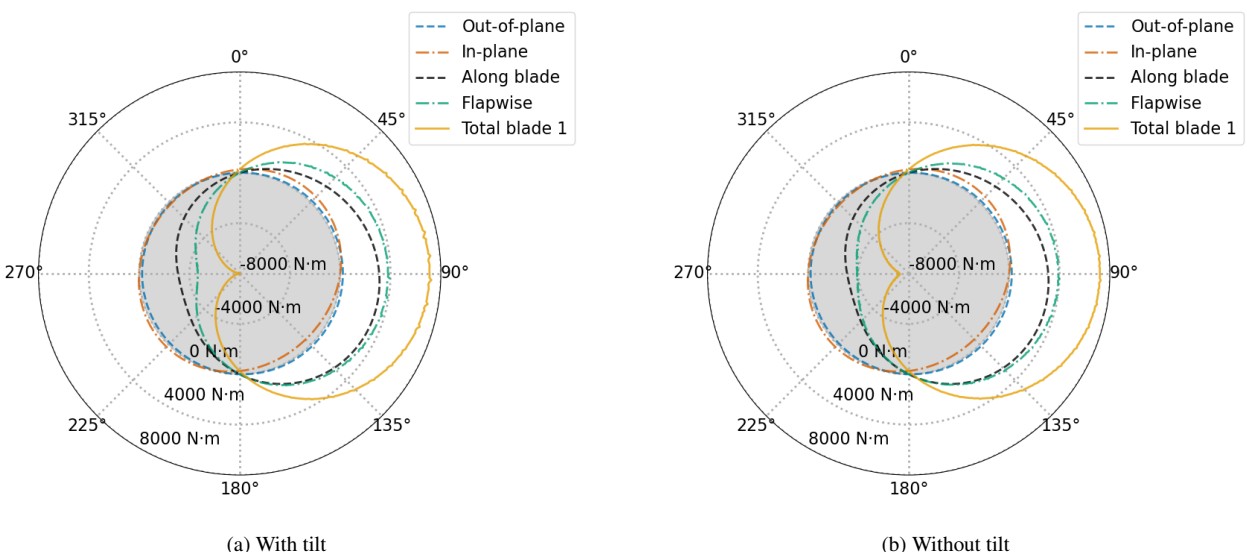

**Figure 5.** Individual blade 1 load contributions to the yaw moment at tower top with respect to azimuth (20 m/s uniform wind speed)

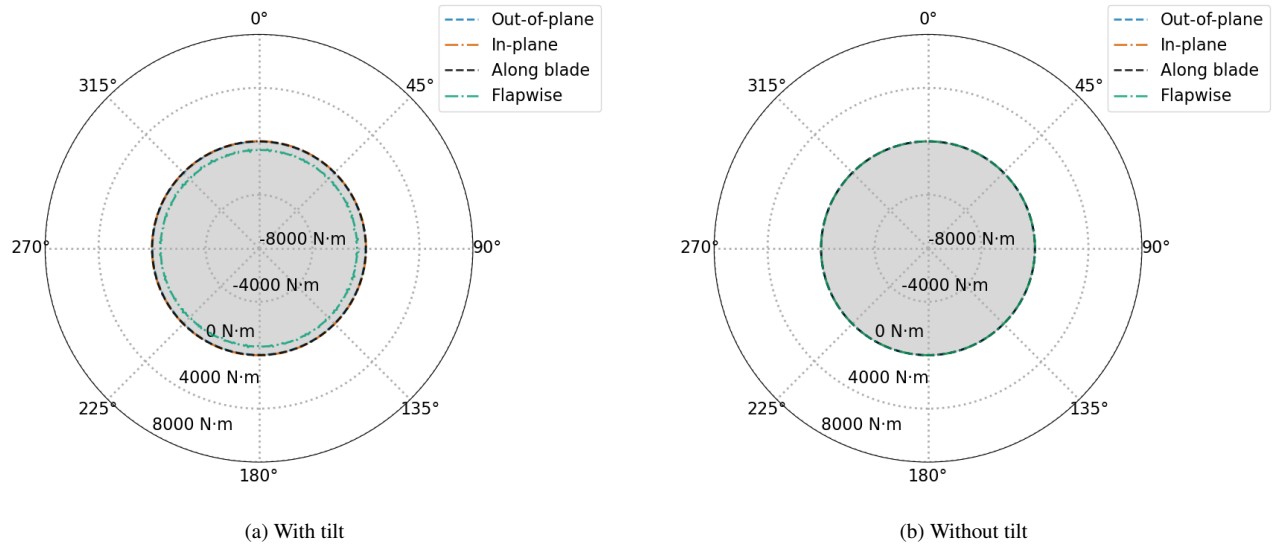

**Figure 6.** Load contributions to the yaw moment at tower top combined for the three blades with respect to azimuth (20 m/s uniform wind speed)

130   from each rotating blade, centrifugal force components along blade and the blade flapwise moment at blade root. All these loads are dependent on the blade azimuthal position.





Figure 5 shows, for blade 1 in the onshore 5 MW NREL wind turbine, the four contributions to yaw moment previously described as a function of the azimuthal position under 20 m/s uniform wind speed, for both with (Fig. 5a) and without tilt cases (Fig. 5b).

135     The most important contributions to the yaw moment come from the blade flapwise moment and the moment caused by the centrifugal along-blade load. Figure 5 shows that they reach values around 4000 N m at 90 º of azimuth and -4000 N m at 270 º, regardless there is tilt or not. On the other hand, the out-of-plane and in-plane moment contributions are very close to 0 N m.

    Figure 6 depicts the same breakdown of load contributions to the yaw moment at tower top as Fig. 5, but each one summed up for the three blades. In the case with tilt (Fig. 6a), all the contributions from the three blades get compensated over the 140  rotor and have a zero value, except for the flapwise contribution, which attains a value of -660 N m and remains constant with azimuth. In the case without tilt (Fig. 6b), the flapwise contribution resulting from the three blades is also zero, and therefore all the curves lie just over the 0 N m circumference. Please note that the centrifugal force components along blade get cancelled when combining the three blades in both the tilt and no-tilt case, because it is assumed there is no mass imbalance in the rotor.

    Accordingly, when all the load contributions are combined for the three blades (Fig. 7), the resulting moment $M_z$ at tower 145  top is -660 N m for the case with tilt (constant magnitude with respect to azimuthal position), as shown in Fig. 7a. In the case without tilt (Fig. 7b), the resulting moment is obviously zero, since all the contributions are zero when summed up for the three blades.

    The above results show the relevance of the shaft tilt in the generation of yaw moment. In the case of onshore and offshore bottom-fixed wind turbines, this yaw moment is absorbed by the foundation. However, in the case of floating turbines these 150  loads can influence the floater response, even if the moment magnitude is relatively small, due to the low stiffness in the platform yaw DoF in certain configurations, particularly in SPM systems. Next section presents the effect of this yaw moment $M_z$ on the response of the DeepCwind OC4 semi-submersible platform using a SPM configuration.

## 4   Effect of the yaw moment from the turbine on the platform dynamics

As has been described in Sect. 3, the rotor induces a vertical moment $M_z$ that can produce a yaw drift of the platform. The 155  amplitude of this platform yaw rotation depends on the stiffness provided by the mooring system. This effect was reported in Jonkman and Musial (2010) for a spar floating platform with a symmetric mooring configuration. In the case of SPM configurations, the yaw stiffness of the mooring system is zero and the effect of yaw moment is particularly critical.

    In the current work, to illustrate and discuss the relevance of this yaw drift, the DeepCWind semi-submersible platform supporting the 5 MW NREL wind turbine is simulated using a SPM system, thus allowing to freely yaw (Fig. 2).

160     Simulations are carried out under NWP steady wind speeds of 8, 12, 16 and 24 m s$^{-1}$ and calm sea (neither waves nor currents). The simulations begin with the initial conditions associated with the steady state response of the FOWT under the same wind speeds. The baseline control strategy (see Sect. 2) has been used in this calculations.



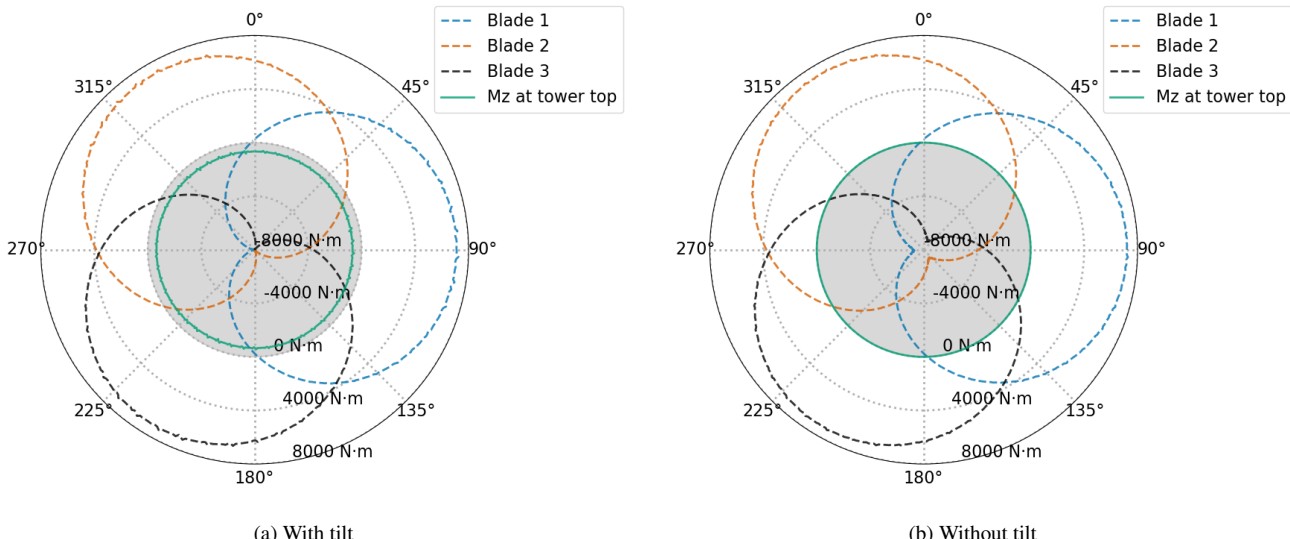

(a) With tilt             (b) Without tilt

**Figure 7.** Tower-top total yaw moment contribution of each blade and corresponding combined contribution of the three blades with respect to azimuth (20 m/s uniform wind speed)

Figure 8 shows the yaw drift of the platform under the different wind speeds. Below rated wind speed, the yaw drift of the platform is close to zero. Conversely, over the rated wind speed, the drift of the platform becomes more relevant. This is consistent with the yaw moments shown in Fig. 4 for a fixed turbine.

At 16 m s$^{-1}$ the drift of the platform stabilises around 39 °. At this yaw position, the destabilising yaw moments generated by the rotor are compensated by restoring yaw moments that appear with the yaw rotation, such as the one due to the weather-vanning effect and the restoring moment that is generated at the rotor under yawed inflow wind, as reported by Wanke et al. (2019).

In order to avoid this yaw drift, the rotor must generate an additional vertical moment that compensates the yaw moments already discussed. This can be achieved with an IPC control strategy described in the next section.

## 5 Individual pitch control strategy to mitigate platform yaw drift

As stated in the introduction, the platform yaw response in upwind SPM-FOWTs is believed to be rectifiable by using IPC strategies. These strategies seem to be the right choice since, by controlling each blade independently, asymmetric moments can be generated in the rotor, which counteract those induced by the turbine (Sect. 3). However, it is still unknown whether an upwind SPM-FOWT is sensitive enough to the moments generated by an IPC strategy. In this section and the following one an answer to this question is provided.



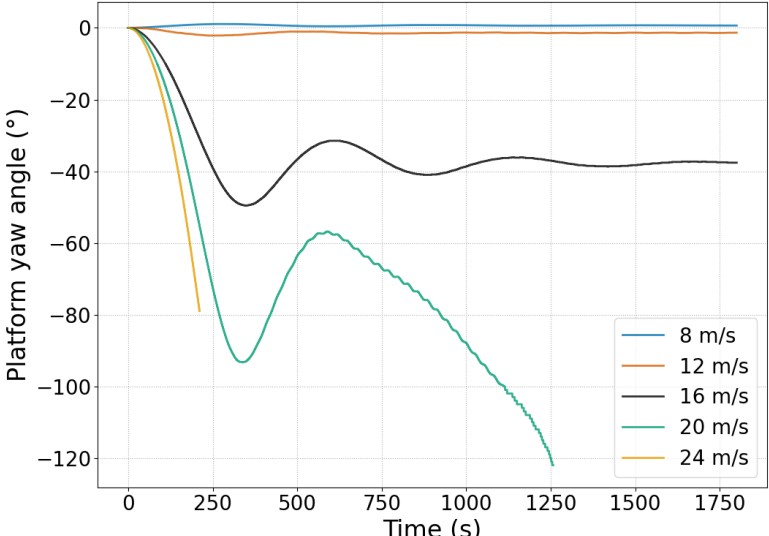

**Figure 8.** Platform yaw drift for different wind speeds (NWP)

The main objective of this IPC loop (yaw-by-IPC) is to keep the platform yaw angle near to zero, i.e. with zero mean and small deviations, in order to maximise power production and reduce structural loads. Nevertheless, the platform yaw angle may not be an available signal in FOWTs; hence misalignment between wind main direction and nacelle angle is used for the control loop, which can be calculated based on the measurement from a wind vane or other similar sensor. This misalignment is directly related to the platform yaw angle, particularly if the nacelle yaw DoF and the tower torsional mode are disregarded. However, there can be some differences, especially at low magnitudes and fast frequencies, due to wind stochastic nature.

In order to reduce the differences between both signals, it is advised to apply a deadband (DB) and a low-pass filter to the misalignment signal (see Fig. 10). The DB used in the current work is based on a hyperbolic tangent function as provided by Navalkar et al. (2014) (Eq. 1):

$$
\phi_{DB} = \begin{cases} -\phi \dfrac{\tanh(\phi+\epsilon)-1}{2} & \text{if } \phi \leq 0 \\[2ex] \phi \dfrac{\tanh(\phi-\epsilon)+1}{2} & \text{if } \phi > 0 \end{cases} \tag{1}
$$

where $\phi_{DB}$ is the resulting DB signal, $\phi$ is the measured raw misalignment and $\epsilon$ is the DB width. The DB signal is then passed through a low-pass filter to remove high frequency misalignments caused by wind.

In Fig. 9 a comparison between raw misalignment, filtered misalignment with DB and platform yaw angle is shown. As expected, the filtered misalignment with DB and the platform yaw angle are quite similar. However, the platform yaw is slower





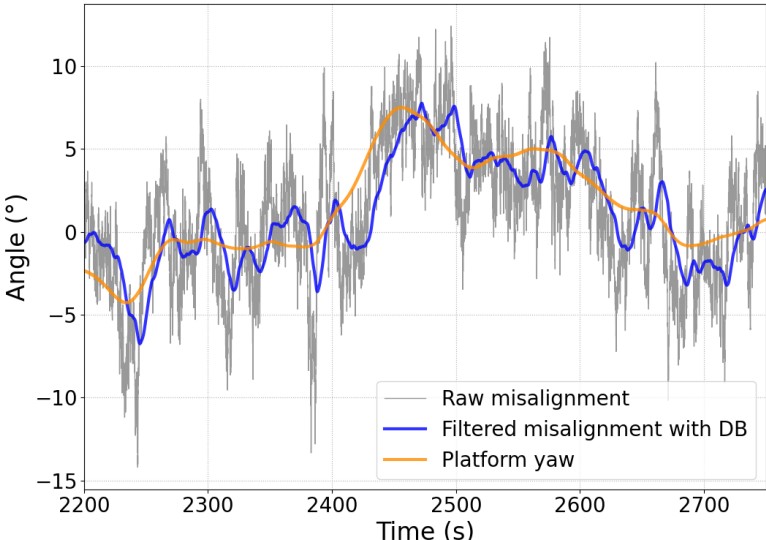

**Figure 9.** Comparison between raw misalignment, filtered misalignment with DB and platform yaw angle

and usually has a phase difference with respect to the misalignment. Reducing the low-pass filter cut-off frequency makes

the filtered misalignment with DB signal slower, but the phase difference increases. Therefore, a trade-off between these two

competing objectives must be found.

The new controller is placed in a feedback loop (Fig. 10), where the measured signal is the misalignment between wind main

direction and nacelle angle. This signal is modified as explained above, then subtracted from the reference value (0 in this case,

as the rotor must be aligned with the wind) and introduced into the controller. The controller output cannot be applied directly

to the blades, as its output is in the non-rotatory frame, whereas the blades are in a rotatory one. To solve this, the commonly

used Inverse Multi-Blade Coordinate (IMBC) Transformation is used (Bossanyi, 2003). For three-bladed wind turbines, the

IMBC is shown in Eq. 2.

$$
\begin{bmatrix} \beta_1 \\ \beta_2 \\ \beta_3 \end{bmatrix} = \begin{bmatrix} \cos(\theta) & \sin(\theta) \\ \cos(\theta + 2\pi/3) & \sin(\theta + 2\pi/3) \\ \cos(\theta + 4\pi/3) & \sin(\theta + 4\pi/3) \end{bmatrix} \begin{bmatrix} \beta_d \\ \beta_q \end{bmatrix}
\tag{2}
$$


where $\theta$ is the azimuth angle, $\beta_n$ is the $n$-th blade pitch angle, $\beta_q$ is the controller output and $\beta_d$ is zero. It is worth mentioning

here that $d$ and $q$ non-rotatory axes correspond to rotor tilt and yaw axes, respectively. That is why, in this IPC case, $\beta_d$ has a

null value and $\beta_q$ is directly the controller output.





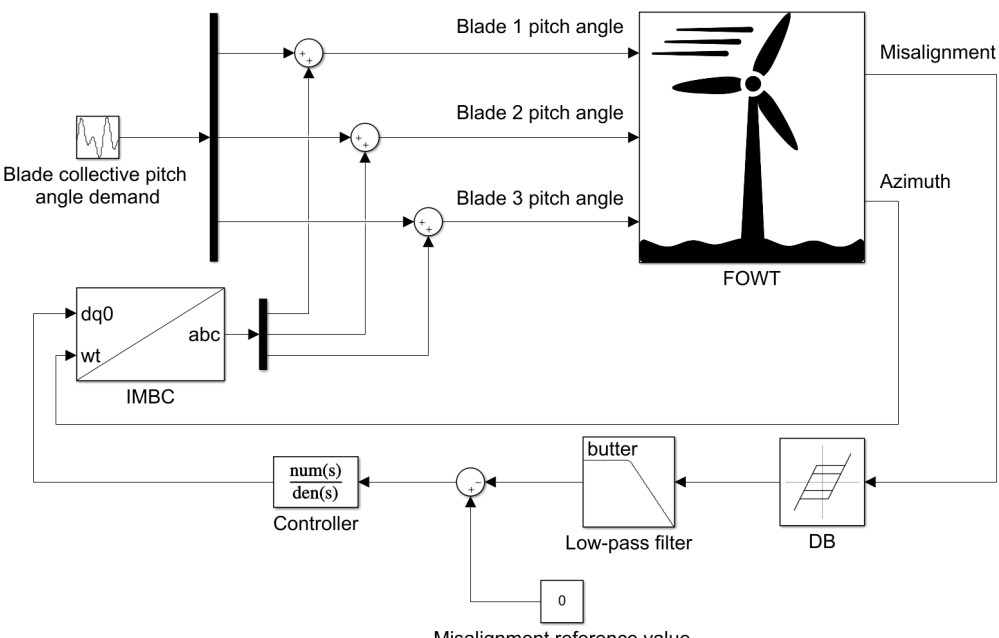

**Figure 10.** Yaw-by-IPC control block diagram

Last, the demanded pitch angle for each blade is added to the collective pitch angle and applied to the corresponding blade.
In summary, a complete block diagram of the used yaw-by-IPC control system is shown in Fig. 10.

To control the misalignment, a conventional Proportional Integral Derivative (PID) controller is used. By adjusting the controller parameters, especially the derivative term, the phase difference between the platform yaw angle and the filtered misalignment can be overcome and a good alignment of the rotor achieved. In the next section results with and without yaw-by-IPC are shown and compared.

## 6 Results

In this section, the effectiveness of the yaw-by-IPC loop strategy is evaluated by means of a set of dynamical simulations of the SPM-FOWT model with shaft tilt in steady NWP and turbulent wind. In both cases main wind direction is aligned with the wind turbine and neither waves nor currents have been considered to allow better interpretation of the results.

### 6.1 Steady NWP wind

In the first case, a wind speed of 20 m s$^{-1}$ NWP with shear of 0.14 is considered. According to the discussion in Sect. 3, for this wind speed the induced yaw moment is quite relevant (Fig. 4) and its effects will be clearly observed. It should be borne in mind that, as wind heading is always 0 °, rotor misalignment and platform yaw are always equal, because there is no nacelle



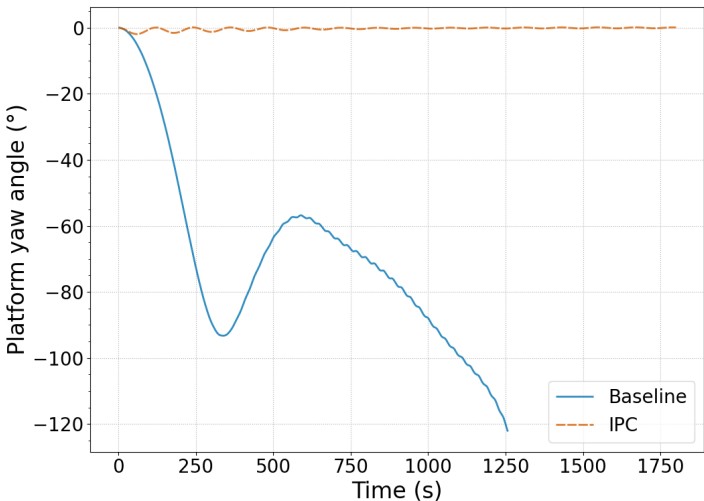

**Figure 11.** Platform yaw angle with and without yaw-by-IPC loop, 20 m s$^{-1}$ NWP wind speed

yaw system active and tower torsional mode is disregarded. Hence, neither the DB nor the low-pass filter explained in Sect. 5 are necessary.

In Fig. 11 it can be observed how the platform yaw is unstable if no yaw drift mitigation strategy is used (baseline controller, see Sect. 2), and it rapidly takes too high values, which would reduce the power production and probably cause the shutdown of the machine. However, using the yaw-by-IPC loop, platform yaw is successfully controlled.

     As well as controlling the platform yaw, regulation of generator speed and power regulation is also appropriately achieved, reaching the rated value (1173.7 rpm and 5 MW, respectively) after the transient period. Without the yaw-by-IPC control these

two variables undergo large variations due to the yaw instability, which are not admissible for a wind turbine. A comparison of the two variables, with and without yaw-by-IPC, can be seen in Fig. 12.

     Besides, other platform DoFs, such as pitch or roll, which are not represented herein for simplicity, are maintained within acceptable levels when the yaw-by-IPC is applied.

     In Fig. 13 the pitch angle of the three blades for the simulated case is plotted looking downwind against the azimuth angle

of blade number 1. With this figure it is possible to see how the IPC varies the pitch angle in one rotation to create a restorative moment, which may result counter-intuitive. The pitch angle cycles have some scattering in the figure because of the initial transient period. The graph can be divided in two halves: the right one, which goes from azimuth 0 º to 180 º and the left one, which goes from 180 º to 360 º. At the transition points between halves (0 º and 180 º points), the pitch angle of blade 1 adopts the value of the collective pitch control (≈17 º). When the FOWT platform presents negative yaw misalignment, like in this

case, this means that the right half of the rotor is placed upwind from the nacelle (is more advanced towards the wind) and the other one, downwind. In order to re-align the rotor, different thrust forces must be generated in each half to create a moment. Thus, a higher thrust force must be applied to the right half (azimuth between 0 º and 180 º), which is achieved by reducing the




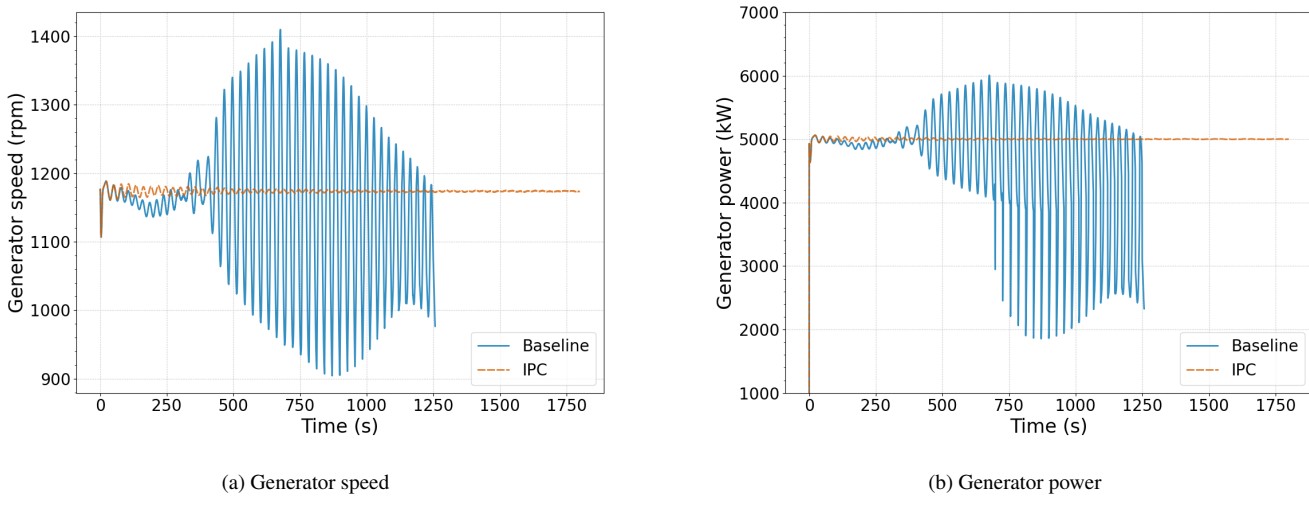

(a) Generator speed        (b) Generator power

**Figure 12.** Generator speed and generator power with and without yaw-by-IPC loop, 20 m s$^{-1}$ NWP wind speed

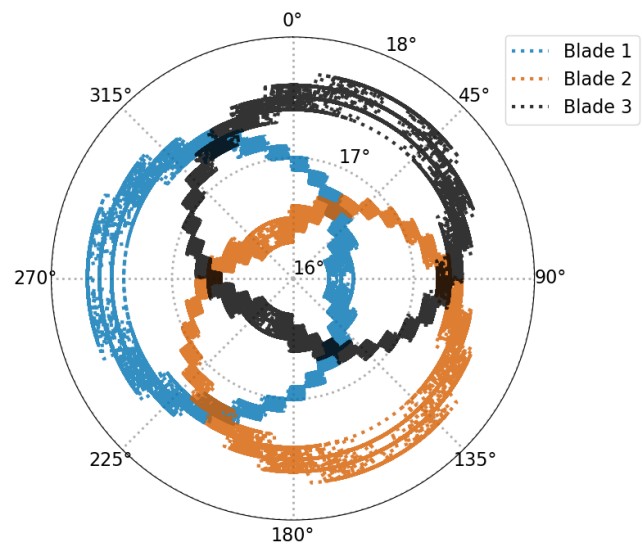

**Figure 13.** Blade pitch angles for yaw-by-IPC with respect to blade azimuthal position, 20 m s$^{-1}$ NWP wind speed

pitch angle of blade 1 in that sector, while it is increased in the left half. This is clearly shown for blade 1 in the figure, and for blades 2 and 3, it is shown with the corresponding phase difference of 120 ° and 240 °, respectively.



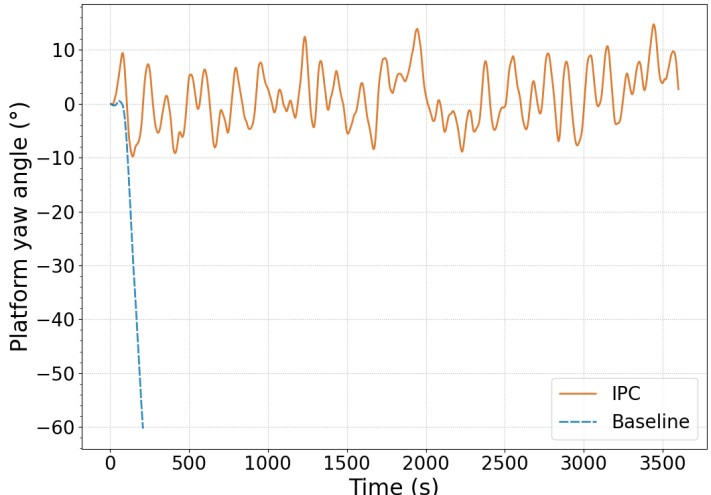

**Figure 14.** Platform yaw angle with and without yaw-by-IPC loop, turbulent wind of 20 m s$^{-1}$ mean wind speed

## 6.2 Turbulent wind

In the second case, a turbulent wind profile is used. Similar to Subsect. 6.1, a mean wind speed of 20 m s$^{-1}$ is selected to clearly showcase the effects under study. A turbulence intensity of 16 % is used, in accordance to values for class A offshore wind turbine in standards (IEC, 2019).

Unlike the case with constant wind speed, now misalignment between rotor and main wind direction is obviously not equal to platform yaw, due to wind stochastic nature. This makes necessary the previously described signal processing (DB and filter) of the measured signal.

As can be seen in Fig. 14, platform yaw angle is successfully controlled. As in the case with steady NWP of 20 m s$^{-1}$, FOWT instability is avoided by mitigating the yaw drift, and platform mean yaw angle is brought near zero. Due to wind turbulence, it is not possible to achieve a constant alignment. However, it is usually maintained within ±10 º and its maximum absolute value never exceeds 15 º, which are considered to be acceptable values. It is worth noticing that the platform yaw response without IPC controller has a drift motion similar to the results observed for the cases of steady winds. Thus, the fluctuation from the turbulent winds does not produce important differences in the platform yaw drift.

Apart from keeping the wind turbine aligned with the wind, the yaw-by-IPC loop does not interfere generator speed and power regulation, similar to the previous steady case. Generator speed and power signals can be observed in Fig. 15. Overspeed and overpower values are kept always below 4 %, which indicates a very tight regulation.

Furthermore, this strategy maintains the rest of platform DoFs within acceptable ranges, as shown in Table 1.





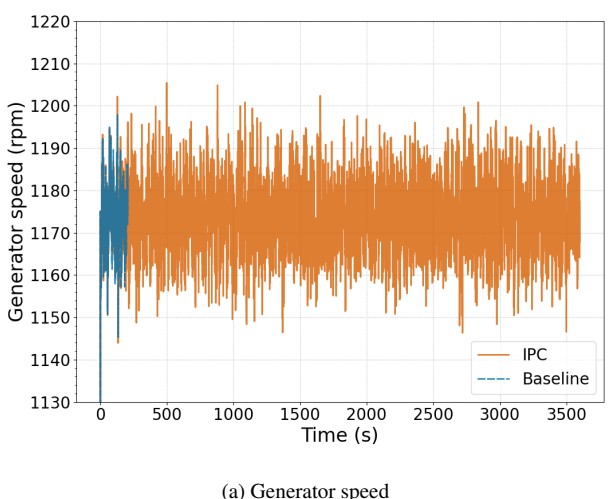
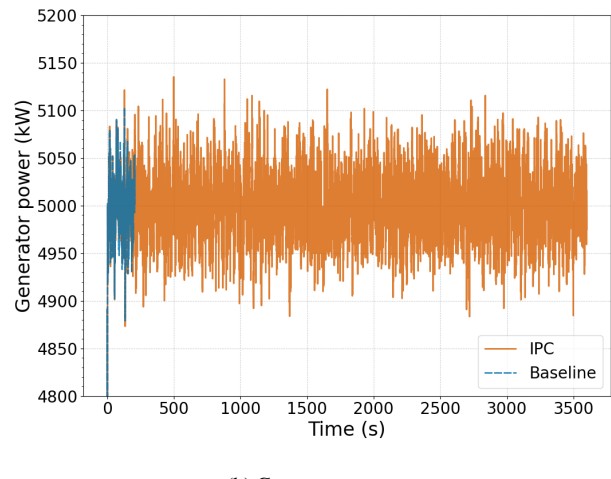

(a) Generator speed        (b) Generator power

**Figure 15.** Generator speed and generator power with and without yaw-by-IPC loop, turbulent wind of 20 m s$^{-1}$ mean wind speed

**Table 1.** Platform DoFs statistical values for turbulent wind of 20 m s$^{-1}$ mean wind speed with yaw-by-IPC

| Platform DoF | Mean | Standard deviation | Maximum | Minimum |
|---|---|---|---|---|
| Surge (m) | 4.9774 | 1.1262 | 7.5379 | 2.0466 |
| Sway (m) | -0.3684 | 1.4400 | 3.3314 | -3.9776 |
| Heave (m) | 0.0053 | 0.0272 | 0.0803 | -0.0899 |
| Roll (°) | 0.3246 | 0.4575 | 1.5285 | -1.0295 |
| Pitch (°) | 1.9680 | 0.7965 | 4.2359 | -0.0606 |
| Yaw (°) | 1.2384 | 5.1460 | 14.7528 | -9.7925 |

The alignment of the FOWT, as well as ensuring a good speed and power regulation, and minimisation of other platform DoFs, is strongly believed to have a positive impact in the mooring line tensions, as in other system loads. Nevertheless, this analysis is out of the scope of this paper and will be carried out in detail in future studies.

All in all, the yaw-by-IPC loop is demonstrated to maintain the upwind FOWT aligned with the main wind direction, while it allows a smooth power and speed regulation, and reduced platform motions.

# 7   Conclusions and future work

This work has presented the relevance of the yaw moment generated by an upwind rotor, on the dynamics of a offshore floating wind turbine with SPM system. This effect tends to misalign the system with respect to the wind direction and can potentially





destabilise it. Additionally, this work has also demonstrated the capability of IPC control strategy based on the nacelle yaw misalignment to mitigate the effect of such moments on the platform yaw drift.

The yaw moment induced by the turbine increases with wind speed, but its magnitude highly depends on the shaft tilt angle. The main contributors to this tilt-related moment are shown to be the projection of the generator torque on the vertical axis and the blades flapwise load, provided there is no rotor mass imbalance.

For onshore and offshore bottom-fixed wind turbines, this tilt-related moment is absorbed by the foundation without further consequences. Conversely, in FOWTs, particularly those with SPM system, the effect of the vertical moment induced by the turbine becomes especially important, due to the lack of stiffness in yaw rotation to counteract it. In that case, the FOWT response results in a platform yaw drift that depends on the magnitude of wind speed and can strongly impact the wind turbine power production and loads.

To avoid this yaw drift, a solution based on an IPC strategy is presented. This control strategy is capable of generating asymmetric moments in the rotor that counteract the destabilising ones and keep the turbine aligned. This IPC strategy, based on yaw misalignment measurements and known as yaw-by-IPC, had already been tested in other turbine configurations in the past, but not for the challenging case of upwind SPM-FOWTs.

Simulations for the SPM DeepCwind OC4 platform supporting the 5 MW NREL wind turbine have shown that the yaw-
by-IPC loop is an adequate strategy to avoid the yaw drift in upwind SPM-FOWTs. At the same time, it does not affect the generator speed and power regulation, and maintains other platform DoFs within acceptable levels.

In future work, more complex and realistic environmental conditions will be analysed and tested, including the simultaneous effect of waves, ocean currents and misaligned wind. This will allow the feasibility assessment under multiple misalignment sources. At control level, the effect on structural loads and mooring line tensions will be assessed in detail.

*Author contributions.*  IS designed the controller, post-processed the simulations, analysed the results and wrote the manuscript. FV analysed the yaw moment produced in both onshore and floating offshore wind turbines, developed the hydrodynamic model and wrote the manuscript. RM developed the aerodynamic model, ran the simulations and reviewed the manuscript. IE and JA reviewed and edited the manuscript.

*Competing interests.*  At least one of the (co-)authors is a member of the editorial board of Wind Energy Science.

*Acknowledgements.*  This work has been conducted within ARCWIND project (Adaptation and implementation of floating wind energy
conversion technology for the Atlantic region (http://www.arcwind.eu/)), which is co-financed by the European Regional Fund through the Interreg Atlantic Area Programme under contract EAPA 344/2016.

The authors also want to thank the Government of Navarre for the funding provided by the "Ayudas para la contratación de doctorandos y doctorandas por empresas y organismos de investigación y difusión de conocimientos: doctorados industriales 2019-2021" program that has been used for the development of AeroVIEW.



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
