# Peer review of "Platform yaw drift in upwind floating wind turbines with single-point-mooring system and its mitigation by individual pitch control"

_Wind Energy Science, 2022_

## Author Response (AR1)

**Answers to Comments**

December 22, 2022

Dear Editor,

This document indicates the comments from RC1 and RC2, our response and where these comments are incorporated in the new version of the document.

**Response to RC1 comments**

*Would it be possible to elaborate on the model used to design this controller? Is it a linearized version of the model or some other method is used to design the controller?*

The PID controller parameters have been tuned using time domain simulation of the full nonlinear model, as the linearised models presented reliability issues.
A comment detailing this question has been indicated in line 222 of the revised paper.

*I also have a question about the controller gains. Results are shown for several cases with different inflow velocities. Is the controller kept the same for all different wind speeds, or is some form of gain scheduling necessary?*

This paper contains only the results with IPC for cases with 20 m/s of wind speed (both constant and turbulent wind), hence the controller is the same one for both cases. However, there are important differences in the FOWT dynamics among wind speeds, as can be seen in Figure 8, for example. We believe a gain scheduling algorithm would be advisable, but a deeper analysis is necessary.
We are currently working on this and a related comment has been included in the "Conclusions and future work" section (line 301).

*Finally, would it be possible to get time graphs for the blade pitch angle? It might be interesting to see the actuation signal over time, to see if this kind of control action can also be realistically applied on a real wind turbine.*

Thank you for the suggestion, a time graph of the three blades pitch angle has been included in the new version of the paper (Figure 12b).

*I also have a question regarding some of the units on Figures 3,4 and 7. The blades are capable of giving up to 8 kNm of torque at 20 m/s, whereas the graphs for moment generated by the turbine are an order of $10^3$ higher magnitude. Is this mislabeling the y -axis, otherwise how can we influence the introduced moments with blade pitch angles?*

Thank you for pointing out this. There was an error in the units of the Figures 3 and 4, that are Nm, instead of kNm. We have fixed this in the new version.

*For Figure 8, when a platform is yawed at high angles (around 90 degrees) w.r.t. inflow conditions, how can it still produce any meaningful thrust to result in a moment that keeps the platform yawing? I find the 20 m/s case interesting, as it continues to yaw even though the turbine is facing away from the wind inflow direction.*

Even at 90$^{\underline{o}}$, there is still some thrust in the rotor, which depends on the blades pitch angle. This thrust is obviously lower than when the FOWT is facing the wind, but it has an impact, due to the lack of yaw stiffness. Furthermore, other aerodynamic effects (explained in the paper) cause the yaw drift, apart from the thrust.

**Response to RC2 comments**

**Introduction**

*Please also mention that yaw misalignment is very important for power capture. This should be the primary goal of a yaw control system and one of the main drawbacks of an SPM system.*

We completely agree, thank you very much for pointing out this.
It has been mentioned in line 23 of the new version of the paper.

*L28: please cite where a free-yawing structure has been shown to reduce structural loads and be more specific about which loads are reduced.*

Many thanks for the comment, the cite [Netzband, 2020] has been included in the revised paper (line 29).

*L48: what is an important yaw response?*

The expression has been removed in the paper, as it could be misleading (line 48).

*L50: please revise this sentence. Although [a] SPM configuration helps to improve the...*

The sentence has been rephrased (line 51).

**Figure 3**

*I recommend using plain English in your legends and describing each case in the caption, so a reader can quickly search for results in the figures and captions.*

Thank you for the suggestion, some legends have been updated and the description of the figures has been included in the captions.

*I think that Figs 3 and 4 are very similar and could be shown side by side for a more interesting result.*

Thank you, it has been done in the new version of the paper.

**Figs 5 and 6**

*Please describe why it is important to look at each blade's individual contribution. These rose plots are not typical in wind energy papers, so some guidance on how to interpret them would be helpful to the reader. What information is this adding, compared to Figs. 3 and 4?*

Figure 3 and 4 present the resulting aggregated rotor moments that have an impact in the global system dynamics, inducing a potential misalignment of the system. These plots present the effect of NWP and tilt on the moments. Figure 5 and Figure 6 provide a better insight of the physics that are causing the effect. Figure 5 shows the contributions of each blade and load component to the aggregated moment. As the contributions depend on the

blade position, they are presented on an azimuthal representation, showing that at $90^o$ ($0^o$) these contributions are maximum (minimum). Figure 6 shows how these contributions cancel totally when tilt angle is 0, but a relatively small moment, constant with azimuth, remains if there is tilt. This moment is responsible for the effect studied in this paper.

A better description of the logic behind these figures has been introduced in line 138 of the revised paper, highlighting their adequacy for azimuth-dependent magnitudes.

**Section 4**

*What are the parameters of the low pass filter? PID gains? More parameters make it easier to repeat the study.*

With this article we just want to demonstrate the ability of this IPC strategy to mitigate the yaw drift, hence specific parameters of the yaw-by-IPC loop are not considered necessary to replicate the study.

*How were the gains tuned? Does the result change with wind speed?*

The controller gains have been tuned using time domain simulation of the full nonlinear model. The system dynamics change with wind speed, hence some nonlinear control algorithm would be advisable, but a deeper analysis is necessary.

An explanation has been added in line 222 of the revised paper.

**Section 6**

*The fact that it works is great, but the comparison shown (especially generator speed/power) is not quantitative. Is there a trade-off between IPC effort (tilt and yaw pitch angle) and yaw regulation/generator power? Near rated, where IPC costs power, is there some optimal effort vs. yaw regulation? Pitch actuation effort can be quantified with pitch travel and the number of direction changes.*

First of all, it must be borne in mind that the yaw drift has to be mitigated to ensure the feasibility of this type of FOWTs. This is the main objective of the paper. Besides, similar to other control problems, there is a trade-off between yaw regulation and IPC effort. In this case, the controller parameters have been tuned so that the platform yaw angle is maintained below $10^o$ most of the time. Near rated wind speeds have not been simulated with the new control loop, but they will be further investigated in the future.

*A more interesting comparison would be with no IPC and the standard mooring configuration: does it have less yaw motion and more power production? If it is nearly the same, then there is a nice argument for the SPM and no yaw actuator.*

We agree. This comparison will be part of future work.

A sentence has been added in the "Conclusions and future work" section (line 305).

*Have you optimized the gains to achieve the best possible yaw regulation? Is there an upper limit on the yaw regulation that can be achieved by IPC? This is the kind of thing I was hoping to learn from this article.*

As explained above, controller parameters have been chosen so that yaw drift is maintained below $10^o$ most of the time, which is thought to be an adequate range. The control is able to achieve tighter yaw limits, at the expense of greater IPC usage. The upper limit on the yaw regulation that can be achieved by IPC will, of course, depend on the turbine and floating platform used, but in general, the results shown in the paper are believed to be suitable for a 5 MW wind turbine.

*Fig 13: I'm not sure this rose plot is the proper way to show these results. Some quantitative measures are provided above. Do the blades only need to vary from 16.5 deg to 17.5 deg? If a higher gain and larger IPC contribution were used, would the yaw motion vary less?*

A temporal graph has been added and explained in the new version of the paper (Figure 12b), in order to ease the understanding of the yaw-by-IPC. The attained yaw variation has been a trade-off with reasonable blade pitch limits and usage.

*Fig 13: L235: why is it counterintuitive?*

The expression has been removed in the paper, as it could be misleading (line 248).

*I'd expect there to be more interesting trade-offs near rated and with misaligned wind/waves. What happens in these cases?*

It has not been analysed in the current work, but will be further examined in the future.
A sentence has been added in the "Conclusions and future work" section (line 303).